# Sensing and Navigation for Multiple Mobile Robots Based on Deep Q-Network

Yanyan Dai , Seokho Yang and Kidong Lee *

Robotics Department, Yeungnam University, Gyeongsan 38541, Republic of Korea; yanyan_dai@ynu.ac.kr (Y.D.); seokhoyang1996@gmail.com (S.Y.)
* Correspondence: kdrhee@yu.ac.kr

**Abstract:** In this paper, a novel DRL algorithm based on a DQN is proposed for multiple mobile robots to find optimized paths. The multiple robots' states are the inputs of the DQN. The DQN estimates the Q-value of the agents' actions. After selecting the action with the maximum Q-value, the multiple robots' actions are calculated and sent to them. Then, the robots will explore the area and detect the obstacles. In the area, there are static obstacles. The robots should detect the static obstacles using a LiDAR sensor. The other moving robots are recognized as dynamic obstacles that need to be avoided. The robots will give feedback on the reward and the robots' new states. A positive reward will be given when a robot successfully arrives at its goal point. If it is in a free space, zero reward will be given. If the robot collides with a static obstacle or other robots or reaches its start point, it will receive a negative reward. Multiple robots explore safe paths to the goals at the same time, in order to improve learning efficiency. If a robot collides with an obstacle or other robots, it will stop and wait for the other robots to complete their exploration tasks. The episode will end when all robots find safe paths to reach their goals or when all of them have collisions. This collaborative behavior can reduce the risk of collisions between robots, enhance overall efficiency, and help avoid multiple robots attempting to navigate through the same unsafe path simultaneously. Moreover, storage space is used to store the optimal safe paths of all robots. Finally, the multi-robots will learn the policy to find the optimized paths to go to the goal points. The goal of the simulations and experiment is to make multiple robots efficiently and safely move to their goal points.

**Keywords:** Deep Q-Network; collaborative behavior; multiple mobile robots; optimal safe path storage space; sensing and navigation



## 1. Introduction

Autonomous navigation is an attractive research topic in the mobile robot field [1]. Simultaneous Localization and Mapping (SLAM) is a frequently used navigation approach [2]. The robot starts to move in an unknown location in an unknown environment; based on information from the sensor during the movement, it localizes itself while building a map, so as to achieve autonomous localization and navigation. However, in order to build a map, SLAM requires the exploration step, which is time-consuming in large environments. Self-driving involves navigating complex environments with various traffic scenarios, pedestrians, and unpredictable events. It is a problem in multi-agent interactions [3]. As discussed in [4,5], Reinforcement Learning (RL) algorithms can handle this complexity by learning from experience and capturing subtle patterns and correlations in the data. In addition, by interacting with its environment and receiving feedback in the form of rewards, an RL algorithm can optimize its policies to make better driving decisions [6,7]. Deep Reinforcement Learning (DRL) has emerged as a promising alternative to addressing the challenges of navigation within dynamic environments, including the limitations of SLAM [8]. Although some researchers try to apply DRL in robot navigation, it is still rarely used in real-world applications [9]. In [1], to train the agent, AGENA2D generates

randomized training environments and provides semantic information. The results have a better safety and robustness performance than when using the traditional dynamic window approach. In [10], self-learning robot navigation is applied to a real robot in an unknown environment without a map or planner. In [11], Deep Q-learning is used for autonomous mobile robot navigation in unknown environments. In [8], a DRL MK-A3C algorithm is proposed to navigate non-holonomic robots with continuous control in an unknown dynamic environment with moving obstacles. However, all of these papers [1,8,10,11] apply DRL to single mobile robot navigation. For multi-robot navigation, there are several challenges. For example, moving robots are also dynamic obstacles. Therefore, all robots are in a dynamic environment. The robots may not obtain the whole observation. Based on [12], multi-agent learning technologies include three parts: multi-agent learning, DRL, and multi-robot applications. Some papers [13–15] test DRL for multi-robot navigation in simulation, but it is hard to reproduce this in the real world. Ref. [16] proposes to use randomization to train policies in simulated environments to reduce the transfer of performance from the simulation to the real world. They communicate with robots to share experiences. Ref. [17] proposes DRL for UAV swarms. It uses an observation history model depending on a CNN and a mean embedding method to solve communication limitations. It is also worth noting that while DRL shows promise, there are still challenges to be addressed, such as sample efficiency, safety guarantees, and generalization to unseen environments. Ongoing research aims to improve the capabilities and robustness of DRL-based navigation methods in dynamic environments. Therefore, in this paper, a novel DRL algorithm is proposed for multiple mobile robots to avoid obstacles and collisions in a dynamic environment. In order to improve learning efficiency, multiple robots process DRL simultaneously. The navigation of multiple mobile robots in real environments is validated in the experiment section.

The traditional tabular form of reinforcement learning has limitations when dealing with a large number of states, such as in the game of Alpha Go. Storing every state and its corresponding Q-value in a table would require a significant amount of memory. In addition, searching through such a large table for each state is time-consuming. Neural networks are well suited for addressing these challenges. Instead of using a table to store the Q-values, neural networks serve as function approximators in reinforcement learning. This is known as DRL [3,18–20]. One of the most famous algorithms in DRL is the Deep Q-Network (DQN) [21–24]. One form of DRL is as follows: The states and actions are treated as inputs to a neural network, which then analyzes and outputs Q-values for the actions. This eliminates the need to store Q-values in a table and instead directly uses the neural network to generate Q-values. Another form of DRL is a value function-based approach commonly used in algorithms like Q-learning. The neural network only takes the state as input and outputs the values for all possible actions. Then, following the principles of Q-learning, the action with the highest value is directly chosen as the next action to be taken. The advantage of this approach is that the neural network can approximate the value relationship between states and actions through learning, without the need to explicitly store a Q-value table. This paper relies on the second form to select actions. The DQN employs a neural network to estimate the Q-value function and trains it using backpropagation. By training the neural network, the DQN learns the complex mapping between states and actions, enabling the efficient modeling of large state spaces. Experience Replay and Fixed Q-target contribute to the immense power of the DQN. Experience Replay allows the DQN to utilize a memory buffer to store past experiences. As Q-learning is an offline learning method, it can learn from both current experiences and past experiences, even including experiences from other agents. Therefore, during each DQN update, we can randomly sample some previous experiences from the memory buffer for learning. This random sampling disrupts the correlation between experiences and makes the neural network updates more efficient. Fixed Q-target is another mechanism that disrupts correlations. By using Fixed Q-target, we employ two neural networks with the same structure but different parameters in the DQN. The neural network used for predicting estimated Q-values has the latest parameters, while the neural network used

for predicting target Q-values is an older version. This approach reduces the correlation between Q-value estimates and target values, stabilizing the training process. In this paper, the multiple robots' states are the inputs of the DQN. The network estimates the Q-value of the agents' actions. During learning, if one robot collides with an obstacle, or with other robots, it will stop and wait for the other robots to complete their exploration tasks. The episode will end when all robots find safe paths to reach their goals or when all of them have collisions. In addition, a storage space is designed to store the optimal safe paths of all robots. At the end of each episode, it is checked whether all robots have found safe paths to their goals. If so, the latest safe path length is compared with the stored path length of the storage space. If the length of the latest safe path is shorter, the path of the storage space will be updated to the latest safe path; if not, the latest safe path will be discarded, and the path of the storage space will remain unchanged.

The contribution of this paper is concluded as follows: (1) A novel DRL algorithm based on a DQN is proposed for multiple mobile robots in a cluttered environment. In the area, there are static obstacles. The other moving robots are recognized as dynamic obstacles that need to be avoided. The multiple robots' states are the inputs of the DQN. The deep network estimates the Q-value of the agents' actions. The action with the maximum Q-value is selected, and the multiple robots' actions are calculated and sent to them. Then, the robots will explore the area and detect the obstacles based on the equipped LiDAR sensors. Multiple robots can share their states based on the master PC. It will give feedback on the reward and the robots' new states. Finally, the multi-robots will learn the policy to find the optimized paths to go to their goal points. The proposed algorithm can be implemented in real multiple robots. (2) We proposed that multiple robots explore safe paths to their goals at the same time, in order to improve learning efficiency. If a robot collides with an obstacle or other robots, it will stop and wait for the other robots to complete their exploration tasks. The episode will end when all robots find safe paths to reach their goals or when all of them have collisions. This collaborative behavior can reduce the risk of collisions between robots, enhance overall efficiency, and help avoid multiple robots attempting to navigate through the same unsafe path simultaneously. (3) A storage space is designed to store the optimal safe paths of all robots. At the end of each episode, compare the latest safe path length with the stored path length of the storage space. If the length of the latest safe path is shorter, the path of the storage space will be updated to the latest safe path, if not, the latest safe path will be discarded. (4) Not only the simulations but also the experiment prove the effectiveness of the proposed algorithm; multiple robots efficiently and safely move to their goal points.

The paper is organized as follows: Section 2 describes the DQN reinforcement learning algorithm for multiple mobile robots; Section 3 presents how to apply the proposed algorithm to a multiple robot simulation; Section 4 demonstrates the efficiency and effectiveness of the proposed algorithm with the experiment.

## 2. Deep Q-Network (DQN) Reinforcement Learning Algorithm for Multiple Mobile Robots

In this section, a novel DRL algorithm is proposed for multiple mobile robots to avoid obstacles and collisions, based on the DQN. To improve learning efficiency, multiple robots explore the area at the same time. The DRL algorithm based on the DQN is shown as Algorithm1. In lines 1–3, initialize the replay memory as $D$, the action value function as $Q$, and the target action value function as $\hat{Q}$. In each episode as shown in line 8, the start points are reset and the goal points are assigned to each robot. The output robots' states are defined as $obs_{nt}(n = 1, 2, 3, \ldots\ldots)$, where $n$ means the robot's number. The environment receives the multiple robots' states. In line 13, stack all robots' states into a single array as $OBS_t$. In lines 14–33, the DQN estimates the Q-value of the agents' actions. Based on the maximum Q-value, multiple actions are calculated and sent to the robots. Each robot obtains one action. The action $a_t$ with max $Q_t$ value is chosen and separated to $a_{nt}$ (n = 1, 2, 3, $\ldots\ldots$). In line 34, the robot will explore the area and detect the obstacles

using an equipped LiDAR sensor. The angle value and distance between the LiDAR and the detection point are obtained. If the distance is less than the defined minimal distance, it is recognized to collide with the obstacle. Based on the angle and distance, the value of points x and y in Cartesian coordinate can be obtained. With the robots' localization information, the coordinates of measurement points x and y in the absolute coordinate system can be calculated. Assuming the obstacle is a regular polygon, the center of the obstacle can be estimated. This information will be used in the DQN reinforcement learning algorithm. In the area, there are static obstacles. The other moving robots are recognized as dynamic obstacles that need to be avoided. The other robots' states are shared by communication. In line 35, the robot will feedback on the reward $r_t$ (2) and the new robot's state. If the robot reaches its goal point, a positive reward will be given. If it is in a free space, zero reward will be given. If the robot collides with an obstacle or the other robots or reaches its start point, it will receive a negative reward. During learning, if a robot collides with an obstacle or another robot, it will stop and wait for the other robots to complete their exploration tasks. The episode will end when all robots find safe paths or when all of them have collisions. Stopping and waiting can reduce the risk of collisions between robots. If multiple robots move simultaneously without proper coordination, they may collide with each other, leading to damage or mission failure. When one robot stops and waits, other robots can continue exploring unknown areas. This collaborative behavior can enhance the overall efficiency and speed of the robot team's exploration. If a robot encounters an obstacle, stopping and waiting allows other robots more time to find a safe path to bypass the obstacle. This helps to avoid multiple robots attempting to navigate through the same unsafe path simultaneously, thereby reducing the risks. Additionally, a storage space is used to store the optimal safe paths of all robots. At the end of each episode, it is checked whether all robots have found safe paths to reach the goal points. If they have, compare the latest safe path length with the length of the path stored in the storage space. If the latest safe path is shorter, the stored path in the storage space is updated with the latest safe path. If it is not shorter, the latest safe path is discarded, and the stored path in the storage space will remain unchanged. Finally, the multi-robots will learn the policy to find the optimized paths to go to the goal points.

For each robot, there are five actions ["up", "down", "left", "right", "stop"]. If there are $n$ robots, then for each step, the number of the actions is $5^n$. For example, if there are two robots, Robot 1 and Robot 2, as in Equation (1), the action can be chosen from a two-dimension array. The array has five columns, indicating Robot 1's actions, and five rows, indicating Robot 2's actions. Therefore, there are 25 actions: up-up, up-down, up-left, up-right, up-stop, down-up, down-down, down-left, down-right, down-stop, left-up, left-down, left-left, left-right, left-stop, right-up, right-down, right-left, right-right, right-stop, stop-up, stop-down, stop-left, stop-right, stop-stop. Concatenate the two-dimensional array to a one-dimensional array; the index number of the element is its action number. Therefore, the action is chosen from the array [0, 1, 2, 3, . . ., 22, 23, 24]. As discussed, if there are three robots, the number of actions is 125. Each action can be selected from a three-dimensional array. Concatenate the three-dimensional array to a one-dimensional array; the index number of the element is its action number, which can be organized as [0, 1, 2, 3, . . ., 122, 123, 124].

$$
\begin{bmatrix}
up-up & up-down & up-left & up-right & up-stop \\
down-up & down-down & down-left & down-right & down-stop \\
left-up & left-down & left-left & left-right & left-stop \\
right-up & right-down & right-left & right-right & right-stop \\
stop-up & stop-down & stop-left & stop-right & stop-stop
\end{bmatrix}
\tag{1}
$$

In each step, if the robot reaches its goal point, it will obtain 1 reward, and if it reaches a free space, it will get 0 reward. However, if it collides with an obstacle or the other robots, or reaches its start point, it will obtain -1 reward. After each robot obtains the reward

$rwd_n(n = 1, 2, 3, \ldots\ldots)$, the total reward $r$, shown in the algorithm, is calculated as in Equation (2) and feedbacked.

$$r = rwd_1 + rwd_2 + rwd_3 + \ldots\ldots \qquad (2)$$

---

**Algorithm 1:** DRL based on DQN with experience replay for multiple mobile robots

---

1 Initialize replay memory $D$ to capacity $N$ = 2000
2 Initialize action-value function $Q$ with random weights $\theta$
3 Initialize target action-value function $\hat{Q}$ with weights $\theta^- = \theta$
4 Initial step parameter to 0, in order to control when to learn
5 Initial storage space array with zero elements
6 Initial the length of the path stored in the storage space to 0
7 **for** number of episodes, **do**:
8    Reset environment and output the multiple robots' states as $obs\_1$, $obs\_2$, $obs\_3$, $\ldots\ldots$
9    Store the robots' states to the temp memory
10   Initial done value to zero
11 **while True**:
12     Render the environment
13     Stack arrays $obs\_1_t$, $obs\_2_t$, $obs\_3_t$, $\ldots\ldots$ into a single array $OBS_t$
14     Enter the choose action function based on parameters $OBS_t$ and $done_t$
15       **if** all robots are exploring **or** have completed all exploration tasks, based on done value:
16         Generate a random number n_rand:
17         **if** n_rand < $\varepsilon_t$:
18           Forward feed the $OBS_t$ and get a $Q_t$ value for all actions
19           Choose the action $a_t$ with max $Q_t$ value
20         **else**:
21           Take random action $a_t$
22         Separate $a_t$ for robot1's action $a_{1t}$, robot2's action $a_{2t}$, robot3's action $a_{3t}$ $\ldots\ldots$
23       **else if** robot1 finishes exploring and the other robots are on process:
24         robot1's action $a_{1t}$ = "stop"
25         robot2's action $a_{2t}$ = choosing random action
26         robot3's action $a_{3t}$ = choosing random action
27         $\ldots\ldots$
28       **else if** robot1 and robot2 finish exploring and the other robots are on process:
29         robot1's action $a_{1t}$ = "stop"
30         robot2's action $a_{2t}$ = "stop"
31         robot3's action $a_{3t}$ = choosing random action
32         $\ldots\ldots$
33       $\ldots\ldots$
34     Execute actions $a_{1t}$, $a_{2t}$, $a_{3t}$, $\ldots\ldots$ to all robots
35     Feedback reward $r_t$ and new robots' states $obs1_{t+1}$, $obs2_{t+1}$, and $obs3_{t+1}$, $\ldots\ldots$
36     Stack arrays $obs\_1_{t+1}$, $obs\_2_{t+1}$, $obs\_3_{t+1}$, $\ldots\ldots$ into a single array $OBS_{t+1}$
37     Store transition ($OBS_t$, $a_t$, $r_t$, $OBS_{t+1}$) in D
38     **if** (step > 200) and (step % 10 == 0): # control the start time and frequency of learning
39       Train the model according to loss:
40         Check to replace target parameters
41         Sample random batch memory ($OBS_j$, $a_j$, $r_j$, $OBS_{j+1}$) from all memory D
42         Obtain the target net $\hat{Q}_{j+1}$ value and eval net $Q_j$
43         Train eval net
44         Gradually increase $\varepsilon$, reducing the randomness of behavior
45     **if** all robots have completed all exploration tasks:
46       **if** all robots arrive at the goal points:
47         **if** the latest safe path length < the length of the path stored in the storage space:
48           Store the latest safe paths to the storage space
49       break
50     step = step + 1

---

In the learning step, two neural networks are built: eval net and target net. For prioritized experiment replay, the absolute temporal difference error (value of $Q_{target\_net}$ − value of $Q_{eval\_net}$) is calculated and stored. If the temporal difference error is large, it means that the prediction accuracy still has a lot of space for improvement, more samples need to be learned, and the priority is high. After training, the absolute value of the new temporal difference error is updated in memory. The target Q-value to train the evaluation network in the algorithm can be described as:

$$Y_j = r_j + \gamma max \; \hat{Q}\left(OBS_{j+1}, a; \theta_j^-\right) \tag{3}$$

where $\gamma = 0.9$ is the reward decay value. Based on the replace target iteration value, every replace target iteration step $\theta^- = \theta$.

The architecture of the Deep Q-Network is shown in Figure 1. It includes one input layer, two hidden layers, and one output layer. The input layer provides inputs for the first layer. The inputs of the first layer are multiple robots' states, such as $[x_1, y_1, x_2, y_2, x_3, y_3, \dots]$. The number of neurons in the first layer depends on the number of robots. In our simulations and experiments, three robots are used, and the first layer is set to have 10 neurons. If there are more robots, more neurons should be used in the first layer. The first layer has 10 neurons. The number of neurons in the second layer is the same as the number of the robots' actions. The output is the action number, which has the max $Q$ value for all robots' actions. ReLU activation functions are used for these two layers. The learning rate is 0.01.

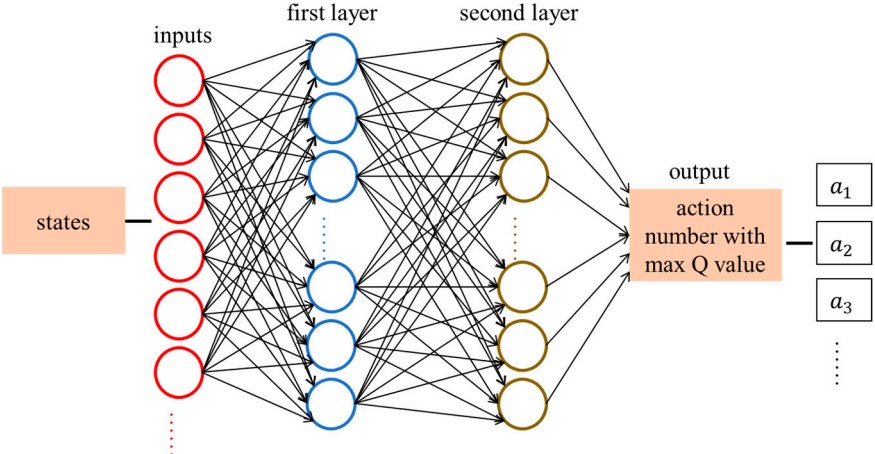

**Figure 1.** The architecture of the Deep Q-Network.

In Figure 1, the action number with max Q-value for all robots is the output, which is discussed in Equation (1). If there are two robots, the action number $a$ can be from the array $[0, 1, 2, \dots, 22, 23, 24]$. Action $a_1$ for Robot 1 and action $a_2$ for Robot 2 are calculated in Equations (4) and (5), respectively.

$$a_1 = remainder \; of \; (a/5) \tag{4}$$

$$a_2 = integer \; of \; (a/5) \tag{5}$$

If there are n robots, the action number $a$ can be from the array $[0, 1, 2, \dots, 5^n - 1]$. Actions $a_{n-2}$, $a_{n-1}$, and $a_n$ are described in Equations (6)–(8), respectively.

$$a_{n-2} = remainder \; of \; \left(remainder \; of \; \left(a/5^{n-1}\right)/5^{n-2}\right) \tag{6}$$

$$a_{n-1} = integer \; of \; \left(remainder \; of \; \left(a/5^{n-1}\right)/5^{n-2}\right) \tag{7}$$

$$a_n = integer \; of \; \left(a/5^{n-1}\right) \tag{8}$$

## 3. Simulations Using the DRL Algorithm

A DQN with a prioritized experience replay RL algorithm was applied to multiple robots. The learning rate was 0.01, reward decay was 0.9, and e-greedy was 0.9. The replacement target iteration was 200. The memory size was set to 2000. The batch size, which is the number of data extracted from the memory at each time, was set to 32. Figure 2 shows the simulation results. The cost function was computed depending on the output Q-value of the eval net and the output Q-value of the target net. The training was completed after 16,000 time steps. The red, blue, and gray squares are the start points of Robot 1, Robot 2, and Robot 3, respectively. The yellow, green, and purple squares are the goal points of Robot 1, Robot 2, and Robot 3, respectively. The black squares are the static obstacles. The red line shows the safe path for Robot 1 from the start point to the goal point. The blue line presents the safe path for Robot 2. It passes the goal point of Robot 1 while tracking the safe path. The gray line is the safe path for Robot 3. It is the shortest path among the three safe paths. Therefore, Robot 3 will arrive at its goal point first and wait for the other two robots to reach their goal points. Figure 3 shows the cost curve. Although the cost decreases, it can be seen that the curve does not decline smoothly. The reason is that the input data in the DQN is changed step by step, and different data will be obtained depending on the learning situation.

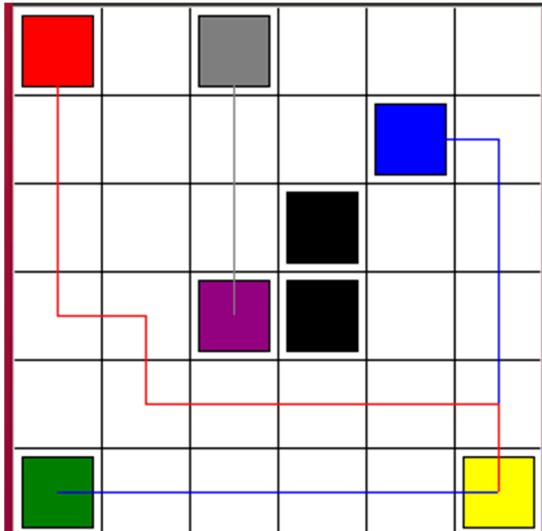

**Figure 2.** Simulation results for Robot 1, Robot 2 and Robot 3, based on proposed algorithms. The red, blue, and gray lines are the safe paths for Robot 1, Robot 2, and Robot 3, respectively.

In order to compare the results shown in Figure 2, the Q-learning algorithm was used to help the robots learn to find safe paths. All environments were set as shown in Figure 2. The start points and goal points were defined as they are in Figure 2. The comparison results are shown in Figure 4. The red line, blue line, and gray line are the safe paths of Robot 1, Robot 2, and Robot 3, respectively. Although all three robots learned to find safe paths, the paths were longer than the paths in Figure 2. After testing several times, sometimes the three robots failed to find safe paths when using Q-learning. However, using the proposed algorithm based on the DQN, the three robots could find the safe paths every time. In addition, the learning time based on Q-learning was longer than the learning time based on the proposed algorithm based on the DQN.

In order to compare the results shown in Figure 2, an A-star algorithm was used for multiple robot navigation. All environments were set as shown in Figure 2. The start points and goal points were defined as they are in Figure 2. The comparison results are shown in Figure 5. Because the A-star algorithm is an off-line navigation algorithm, the static obstacles were assigned to the algorithm in advance. However, the dynamic collision condition among robots could not be considered. Therefore, as shown in Figure 5, there are

collision risks in the circle areas. Based on the A-star algorithm, the red line, blue line, and gray line are the paths of robot 1, robot 2, and robot 3, respectively.

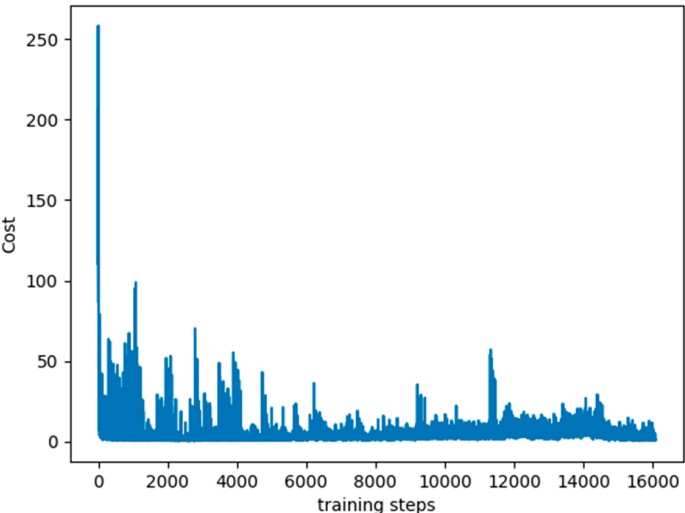

**Figure 3.** Cost curve.

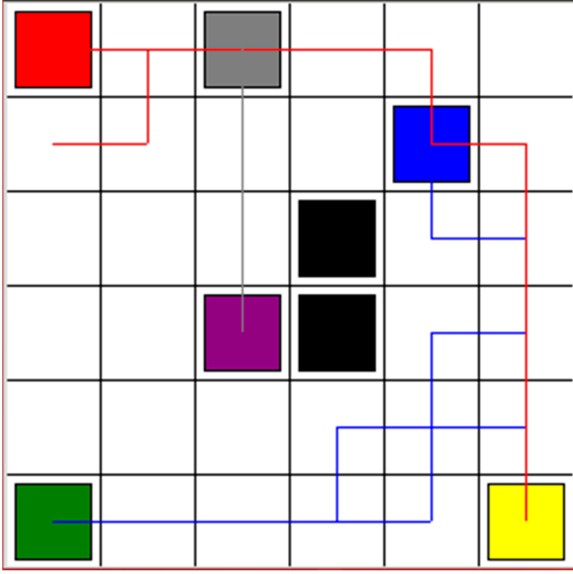

**Figure 4.** Simulation results for Robot 1, Robot 2 and Robot 3, based on Q-learning algorithm. The red, blue, and gray lines are the safe paths for Robot 1, Robot 2, and Robot 3, respectively.

In order to comprehensively prove the effectiveness of the proposed algorithms, the multi-robots were tested in environments other than that shown in Figure 2. As shown in Figure 6, the size of the environment was 10 m × 10 m, which is bigger than the environment in Figure 2. More obstacles were added. Four robots were tasked with finding safe paths. The goal points of the robots were changed. The black squares are the static obstacles. The red line shows the safe path for Robot 1 from the red start point to the yellow goal point. The blue line presents the safe path for Robot 2 from the blue start point to the green goal point. The gray line is the safe path for Robot 3 from the gray start point to the purple goal point. The orange line shows the safe path for Robot 4 from the orange start point to the brown goal point. In the DQN, twenty neurons were used in the first layer and five neurons were used in the second layer. The learning rate was 0.001, reward decay was 0.9, and e-greedy was 0.9. The replace target iteration was 200. The memory size was set to 2000. The batch size was set to 32. Figure 7 shows the cost curve.

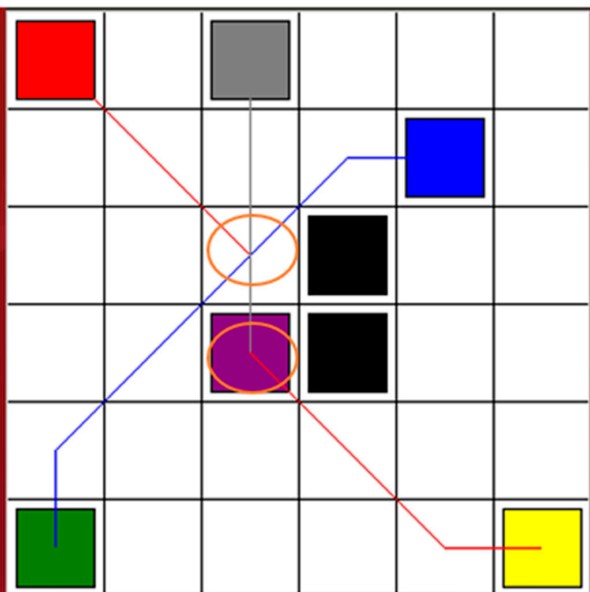

**Figure 5.** Simulation results for Robot 1, Robot 2 and Robot 3, based on A-star algorithm. The red, blue, and gray lines are the paths for Robot 1, Robot 2, and Robot 3, respectively. There are collision risks in the circle areas.

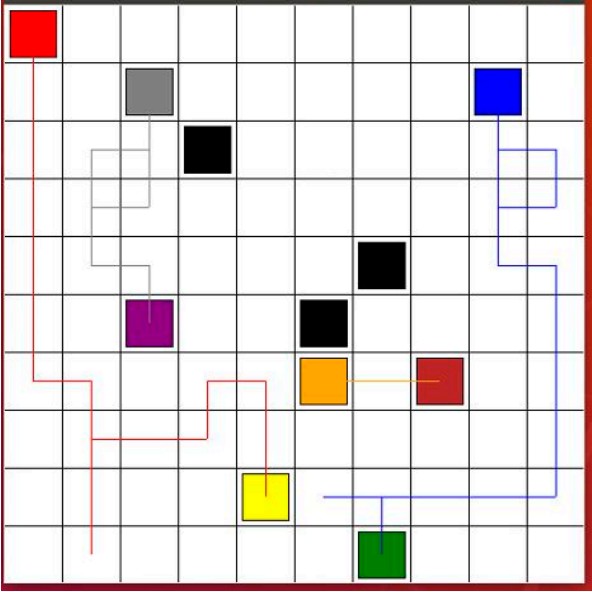

**Figure 6.** Simulation environments for Robot 1, Robot 2, Robot 3, and Robot 4.

The Gazebo Simulator was used to evaluate the proposed algorithm for the navigation of multiple robots. The environment for this Gazebo simulation was the same as the environment shown in Figure 2. The robot model is a LIMO robot, with onboard LiDAR and camera sensors from the company Agilex Robotics. As shown in Figure 8, three robots were added to the simulator. The gray boxes were the obstacles. The size of the GUI cell was $1 \times 1$ m. Firstly, the goal point of each robot was assigned to the robot. Secondly, based on the proposed DQN RL algorithm, the safe path was explored. Finally, the robot would move and track the safe path. The robot subscribed to the "/odom" topic. It helped to obtain the robot's $x$ and $y$ position and calculate the robot's yaw $\theta_{curr}$ value. The robot published the "/cmd_vel" topic and used the linear and angular velocities to let the robot move. If the robot chose the "up", "left", and "right" action, linear velocity was fixed at 0.02 m/s. Angular velocity was calculated based on Equation (9). If the robot selected the "down"

action, linear velocity was −0.02 m/s. Angular velocity was calculated as in Equation (9). If the robot chose the "stop" action, both linear and angular velocities were zero. The safe path exploration and tracking process is shown in Figure 9. At t = 0, the robots start to explore the safe path using the proposed DRL algorithm. At t = 30 s to 60 s, the robots track the safe path. At t = 90 s, Robot 3 arrives at the goal point. At t = 120 s to 240 s, Robot 1 and Robot 2 track their respective safe paths. Finally, they reach their goal points. The robots use the LiDAR sensor to detect the obstacles and the other robots. The video can be viewed at the following link: https://www.dropbox.com/s/wnipjux0yt0s80r/limo_gazebo1.mp4?dl=0 (accessed on 6 June 2023).

$$angular\ velocity = (\text{atan2}((y_w(t+1) - y(t)), (x_w(t+1) - x(t))) - \theta_{curr}(t))/2 \qquad (9)$$

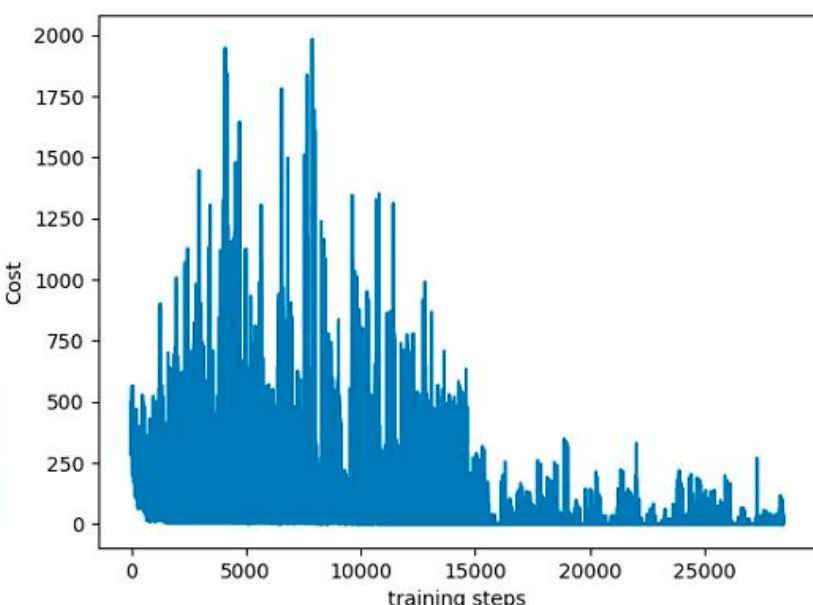

**Figure 7.** Cost curve.

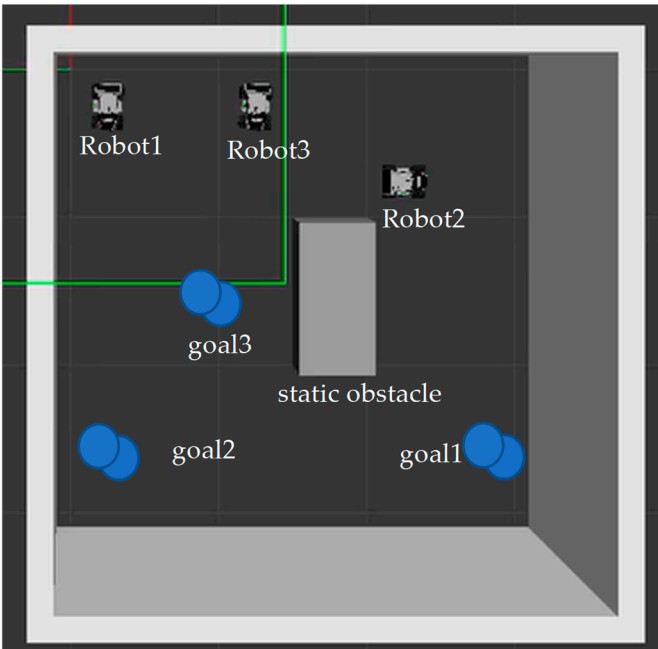

**Figure 8.** Robots in the Gazebo Simulator.

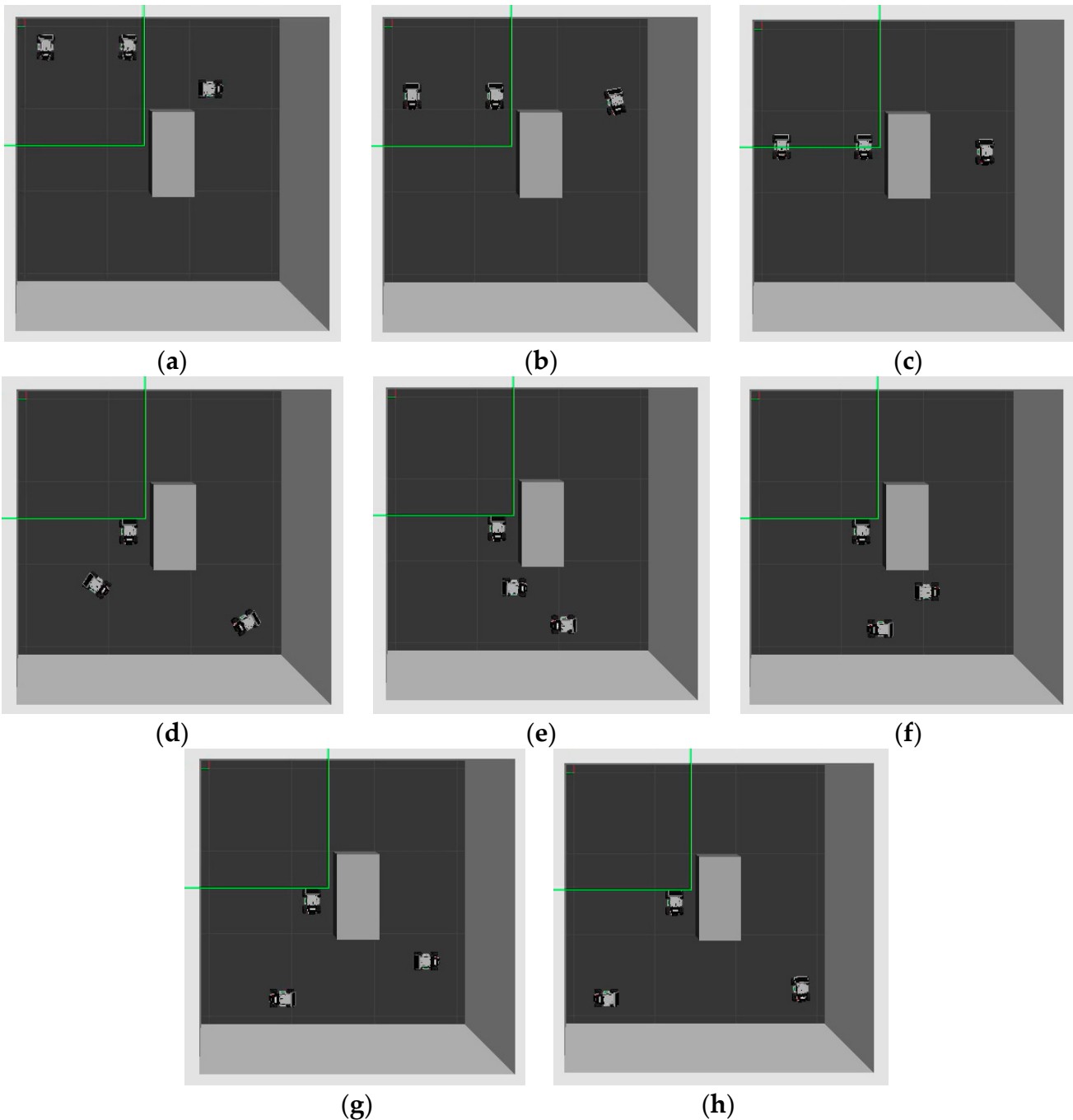

**Figure 9.** Multiple robots explore and track the safe path. (**a**) At t = 0, the robots start to explore the safe paths using the proposed DRL algorithm. (**b**,**c**) At t = 30 s to 60 s, the robots track the safe path. (**d**) At t = 90 s, Robot 3 arrives at the goal point. (**e**–**h**) At t = 120 s to 240 s, Robot 1 and Robot 2 track their respective safe paths. Finally, they reach the goal points.

The robots' trajectories from the start point to the goal point using the DQN are shown in Figure 10. Each robot had a 2D LiDAR sensor to detect obstacles and other robots. The blue line is the trajectory of Robot 1. The dotted line represents the trajectory of Robot 2 and the dashed line is the trajectory of Robot 3.

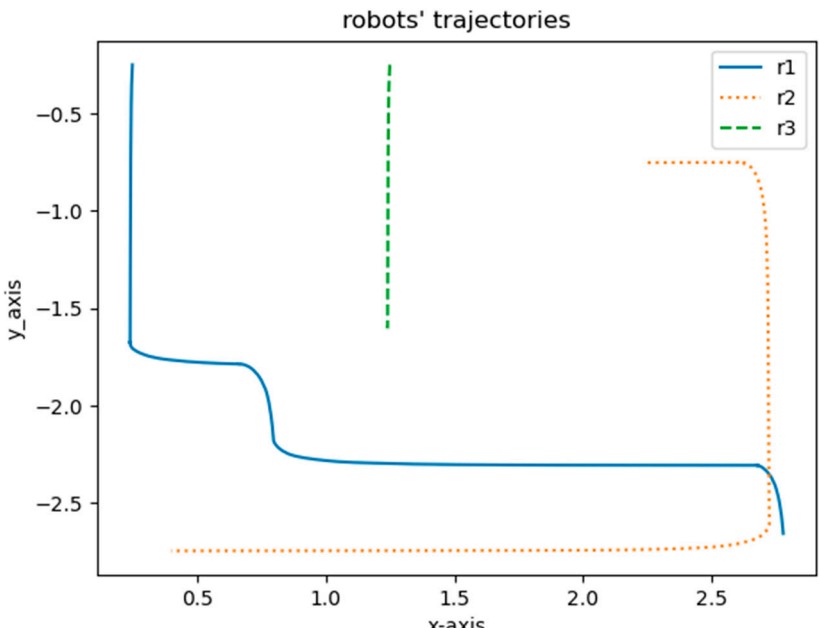

**Figure 10.** Robots' trajectories.

## 4. Discussion of Experiment Using the DRL Algorithm

An experiment was performed to show the effectiveness of the proposed DRL algorithm using three LIMO robots, as shown in Figure 11. Each robot had an onboard MPU 6050 IMU sensor, EAI 2D LiDAR sensor, and stereo camera. The communication interface was WIFI. The controller was NVIDIA Jetson Nano (4G). The operating system was Ubuntu 18.04. A four-wheel differential steering mode was selected to conduct the experiment. The system architecture is shown in Figure 12. It had one master and three slave robots. The master PC and robots could communicate with each other. The master desktop processed the master control node. It obtained the environment and the state of the robots, processed the DRL algorithm, and sent velocity commands to each robot. It obtained environment information using the "/scan" topics from each robot. Based on the LaserScan message, the angle value and distance between the LiDAR and the detection point were obtained. Based on the angle and distance, the value of points x and y in Cartesian coordinate were obtained. With the robot's localization information, each measurement coordinate of points x and y in the absolute coordinate system was calculated. It subscribed to the robots' state topics, including the "/odom" and the "/imu" topics. The robots' moving direction was the x-axis, based on right-handed coordinates. IMU data were used to obtain the heading angle. TF was used to transfer the local odometry coordinate to the global coordinate. Then, the DRL algorithm was processed and the safe waypoints were evaluated. The learning rate was 0.01, reward decay was 0.9, and e-greedy was 0.9. The replacement target iteration was 200. The memory size was set to 2000. The batch size was set to 32. Then, linear and angular velocities were calculated and published to each robot via the topic "/cmd_vel". Each robot was a slave robot. It processed the robot control node. The robots detected the environment and published it in the LiDAR "/scan" topic. The robots subscribed to the "/cmd_vel" topic and tracked the safe path. They also published feedback on their states in the "/odom" and "/imu" topics.

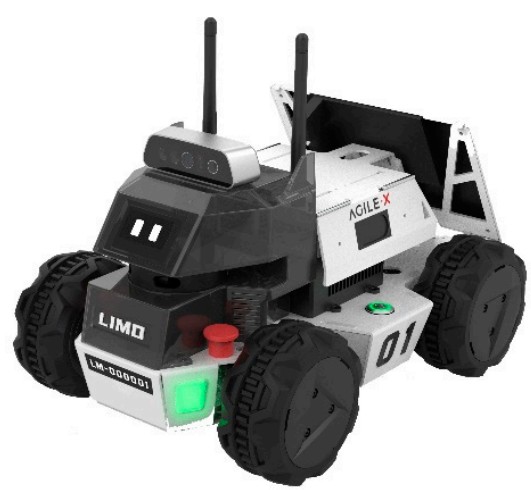

**Figure 11.** LIMO robot.

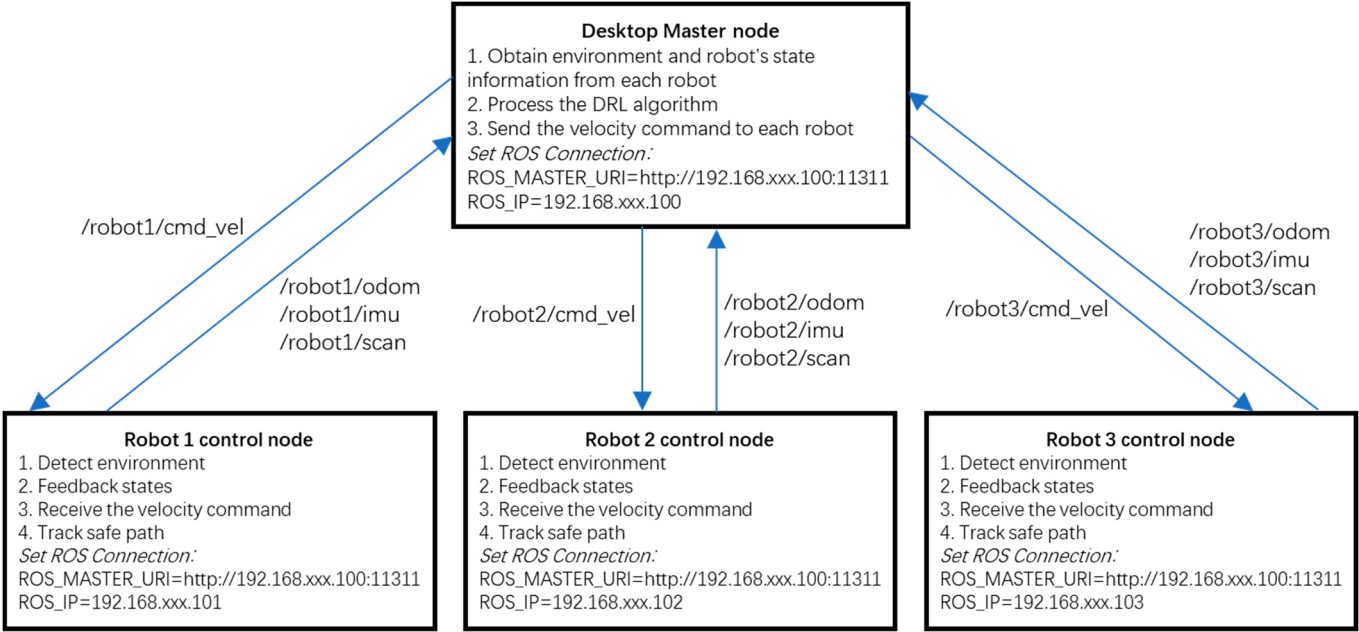

**Figure 12.** System architecture.

The initial points and goal points for the three robots are defined in Figure 2. If the robots chose the "up", "left", and "right" actions, linear velocity was fixed at 0.04 m/s. If the robots selected the "down" action, linear velocity was −0.04 m/s. Angular velocity was calculated based on Equation (9). If the robots chose the "stop" action, both linear and angular velocities were zero. Three robots tracked their goal points and were tested as shown in Figure 13. The blue boxes are the obstacles. If the distance between the robot and the obstacle was greater than 0.1 m, it was recognized as obstacle-free. If the distance between the robot and the goal point was smaller than 0.2 m, it was recognized that the robot arrived at its goal point. In the end, three robots almost reached their goal points. However, during the experiment, slipping errors occurred, especially when the robots turned. A communication delay issue also affected the robots' performance. A video recording of the experiment can be viewed at the following link: https://www.dropbox.com/s/bdyj3hos2nw65mg/experiment_multiple_RL.mp4?dl=0 (accessed on 16 June 2023).

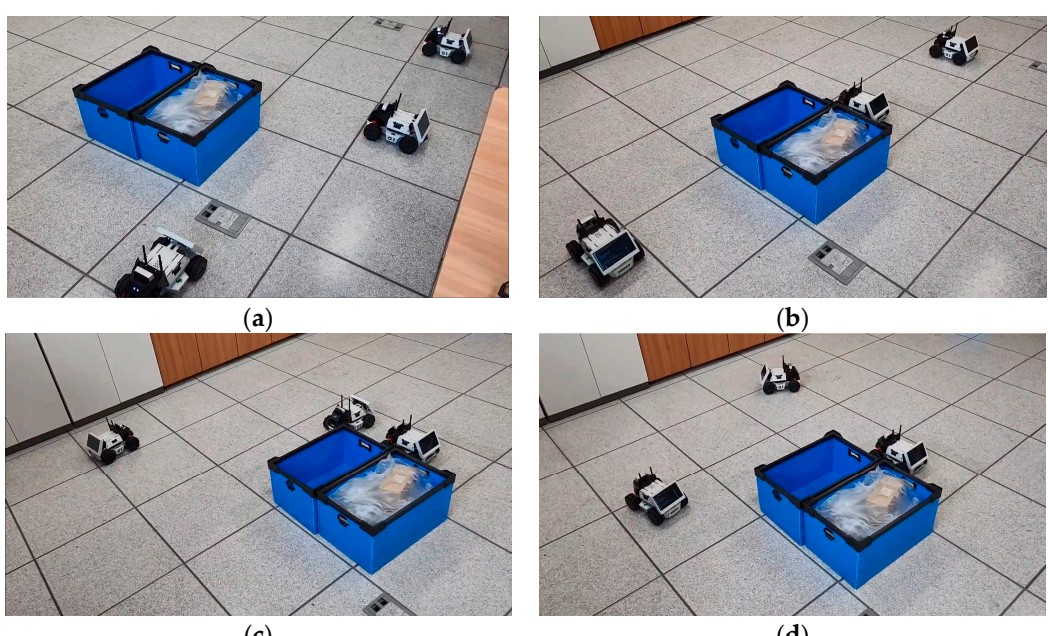

**Figure 13.** Three robots tracking their goal points. (**a**) At t = 0, robots are located at their initial points. (**b**) Robot 3 arrives at the goal point. (**c**) Robot 1 and Robot 2 track their safe paths. (**d**) Robot 1 and Robot 2 almost reach their goal points.

## 5. Conclusions

In this paper, a novel DRL algorithm is proposed to help multiple robots explore and track optimal safe paths to their goal points in a cluttered environment. The DQN estimates the Q-value of the agents' actions. In order to increase exploring efficiency, multiple robots explore safe paths to their goals at the same time. Robots will stop and wait for the other robots to complete their exploration tasks when they collide with an obstacle or other robots. The episode will end when all robots find safe paths to their goal points or when all of them have collisions. This collaborative behavior of multiple robots can reduce the risk of collisions between robots, enhance overall efficiency, and help avoid multiple robots attempting to navigate through the same unsafe path simultaneously. In addition, the storage space is designed to store the optimal safe paths of all robots. Finally, simulations and experiments prove the effectiveness and efficiency of the proposed DRL algorithm.

**Author Contributions:** Conceptualization, Y.D. and K.L.; methodology, Y.D.; software, Y.D.; validation, Y.D. and K.L.; formal analysis, Y.D.; investigation, Y.D.; resources, Y.D.; data curation, Y.D.; writing—original draft preparation, Y.D.; writing—review and editing, Y.D.; visualization, Y.D.; supervision, K.L.; project administration, S.Y. All authors have read and agreed to the published version of the manuscript.

**Funding:** This paper was supported by Korea Institute for Advancement of Technology (KIAT) grant funded by the Korea Government (MOTID) (P0008473, HRD Program for Industrial Innovation).

**Data Availability Statement:** Not applicable.

**Conflicts of Interest:** The authors declare no conflict of interest.

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
