# Peer review of "Sensing and Navigation for Multiple Mobile Robots Based on Deep Q-Network"

_remotesensing, doi:10.3390/rs15194757_

Round 1
Reviewer 1 Report
The manuscript uses a DRL algorithm for multiple mobile robots to avoid obstacles and collisions. There are some issues that the authors should address so that their manuscript can be considered for publication.
- The authors must put more effort in describing DRL in mobile robots and multi-robot systems. The authors must clearly and concisely present the contributions of their paper and the difference with respect to recent and similar papers in the state of the art.
- The authors should perform simulations or experiments with a higher number of robots simultaneously moving.
- The authors should try to present the algorithm in a clearer way so that the reader can have a clearer idea of the method.
Reviewer 2 Report
A new DRL algorithm is suggested in this paper to help multiple robots in exploring and tracking the most safe paths to their goals in cluttered environments. The Q value of the agents' activities is calculated by the DQN. Multiple robots simultaneously explore safe paths to the goals in order to maximize the efficiency of exploration. Overall, the paper sounds good. However, I have some comments as follows.
- The authors should present the process time to run the algorithms.
- The author missed detail information in the experiment part. What is the system architecture? How does the robot localize?
- The simulation and experiment were done in simple environment. More simulations or experiments are required to test the policy.
Reviewer 3 Report
The paper describes a method of using Deep Learning and Reinforcement learning for path planning for multiple robots to perform exploration and obstacle avoidance.
The paper demonstrates that the proposed method works quite well but there are a few issues that need to addressed before I can recommend this paper for publication:
1. The paper should provide information on the observations that are provided to the deep network to determine the action. Is this the absolute positions of the obstacles and robots are is it the relative positions of the robots and obstacles to the robots?
2. The paper indicates that it using a deep learning network to perform the navigation task however the network shown in Figure 1 only has three layers. Most deep learning systems differentiate their neural networks from conventional learning neural networks because deep learning networks have many more layers than three. Can the authors indicate why their networks are deep learning networks and not more conventional neural networks?
3. The paper shows that their proposed network is able to properly navigate the robots. However, there are other algorithms that have been previously proposed to perform this task. It would be best if the authors compared their algorithm with previously proposed algorithms to show the superiority of their algorithm.
If these questions are answered, I think that this paper should be published.
Round 2
Reviewer 3 Report
I think that the proposed new method for robot navigation are useful and give good performance. However, the comparison with prior work should be stronger. There are many other algorithms such as using A-star algorithms optimization searches and other algorithms which have been proposed. Some better descriptions of previous algorithms and some comparisons with these methods would be useful. I think it is very likely that these algorithm comparisons would be in favour of the new algorithm for some useful configurations.
